# A Study of Some Mechanical Properties of Composite Materials with a Dammar-Based Hybrid Matrix and Reinforced by Waste Paper

**DOI:** 10.3390/polym12081688

**Published:** 2020-07-29

**Authors:** Marius Marinel Stănescu, Dumitru Bolcu

**Affiliations:** Department of Mechanics, University of Craiova, 165 Calea Bucureşti, 200620 Craiova, Romania; mamas1967@gmail.com

**Keywords:** hybrid resin, waste paper, composite materials, mechanical properties, chemical structure

## Abstract

When obtaining environment-friendly hybrid resins made of a blend of Dammar natural resin, in a prevailing volume ratio, with epoxy resin, it is challenging to find alternatives for synthetic resins. Composite materials reinforced with waste paper and matrix made of epoxy resin or hybrid resin with a volume ratio of 60%, 70% and 80% Dammar were studied. All samples obtained have been submitted to tensile tests and Scanning Electron Microscopy (SEM) analysis. The tensile response, tensile strength, modulus of elasticity, elongation at break and the analysis of the fracture surface were determined. The damping properties of vibrations of bars in hybrid resins and in the composite materials under study were also examined. The mechanical properties of the four types of resins and of the composite materials were compared. The chemical composition for a hybrid resin specimen were obtained using the Fourier Transformed Infrared Spectroscopy (FTIR) and Energy, Dispersive X-ray Spectrometry (EDS) analyzes.

## 1. Introduction

The last decades have seen an increased use of composite materials, whose components, matrix and/or reinforcing agents come from nature. These new types of materials are called “green composites”, or “bio-composites”. The main advantages of using green composites are: both the fibers and the resins are bio-degradable and compostable; they can be manufactured at a low cost (compared to synthetic composites) because they are abundantly made by nature; they have relatively good mechanical properties. Most “bio-composite” materials that have been studied so far related to natural fibers as reinforcing agents in combination with thermoplastic matrices (polypropylene, polyethylene and polyvinyl chloride), or thermorigid matrices (phenolic, epoxy and polyester resins) (see for example [1]).

The first purpose of this article is to study the properties of composite materials with reinforcement from paper waste (containing cellulose obtained from natural fibers). The most used natural fibers are wood, kendir, flax, sisal and hemp, although there are others that are appropriate for bio composites as well (see [2,3,4,5]). The natural fibers are easier to manipulate and require less energy during the processing, compared to glass or carbon fibers (see [6,7]). From a technical point of view, natural vegetal fibers are already cellulose-based polymer composites (see [8]). The specific tensile strength and toughness make natural fibers be an alternative to traditional reinforcements, such as glass fibers or other filling materials. However, the mechanical properties of natural fibers are usually much worse than traditional reinforcing materials, in general, even to the glass fiber. One justification would be that the natural fibers are not homogeneous, they have non-uniformities both from a dimensional and structural point of view. The properties of natural fibers are influenced not only by natural factors (climatic conditions, soil, seed varieties, etc.), but also by plant processing conditions.

Biomass represents one of the most important resources for humans. Exploiting and using this natural resource plays an important role in sustainable development and environment protection because, as it exists abundantly, it is biodegradable and can be repeatedly renewed. Out of all biomass resources, cellulose, with a yearly production of around 1.5 trillion tons, is the most abundant polymer on Earth and it is considered an almost inexhaustible raw material that can cater to an increasing demand for bio-compatible and environment-friendly products (see [9,10,11]).

Moreover, cellulose displays good mechanical properties, therefore it is useful for the manufacture of composite materials. Thus, the elasticity modulus of cellulose can reach up to 138 GPa in the direction parallel with the chain axis, which is significantly higher than in the case of aluminium (70 GPa) and glass fibers (76 GPa) and it is compatible with high-performance aromatic polyamides (156 GPa, Kevlar and Twaron) and aromatic polyesters (126 GPa, Vectran; 130 GPa, Ekonol) (see [12,13,14,15,16]). The final tensile strength of cellulose I is estimated to be 17.8 GPa, which is seven times higher than that of steel (see [17]). In addition, cellulose I has the highest known specific tensile strength of natural polymers (667 MPa·cm${}^{3}\cdot$g${}^{-1}$) (see [18]). Nevertheless, the mechanical properties of cellulose-based anisotropic composites heavily depend on the dissolution and regeneration conditions. The longitudinal tensile strength decreases as the dissolution time increases due to a decrease in the transversal section in the area of the cellulose fibers which take over the stress and, consequently, there is a reduction of the fraction of the volume fibers. Instead, the transverse tensile strength increases as the proportion of the matrix increased and the composite mixture becomes more homogeneous.

In the case of cellulose-based composites the crux lies in the adherence between fibers and the matrix. This flaw in bio-composites can be adjusted by employing cellulose materials that are chemically similar or identical both to the matrix and to the hardener. Thus, all-cellulose composites (ACC) are mono component-cellulose composites, where the matrix is the regenerated cellulose and the hardening phase generally consists of high-resistance cellulose fibers. This type of composite shows good interface compatibility, bio-compatibility and biodegradability because they have the same cellulose composition of the matrix and the reinforcing phase. The ACC viscoelastic properties are determined by the viscoelastic properties of the cellulose and allomorph chains. The amorphous cellulose shows viscoelastic behavior that involves different molecule movements according to temperature (see [19,20]).

The second purpose of this article is to obtain “environmentally friendly” composite materials whose matrix is a hybrid resin with the majority component of natural resin. The first thermo-rigid-biological matrices were obtained from vegetable oils (bio-degradable), not requiring a polymerization process (see [21,22,23]). Sandarac, Copal and Dammar are the most used vegetable resins. These natural resins are insoluble in water but are slightly soluble in oil, alcohol, turpentine and, partially, petrol. They form, together with certain organic solvents, solutions used as covering varnishes (see [24,25,26,27]). One shortcoming of natural lacquers is that they cannot form thick resins (see, for example [24,25]). A solution for eliminating this shortcoming would be the use of a hybrid resin, obtained by combining several components, with at least one of them being organic and at least another one synthetic (see [28]). Hybrid resins are an environmentally-friendly alternative compared to the synthetic resins. Comprehensive investigations into the hybridization of resins and fibers are available in [29,30]. The studies on these resins focused more on their chemical composition and properties, presented in [31,32], and less on their mechanical properties.

Dammar, the most commonly used natural resin, is harvested from the Dipterocarpaceae family trees of India and East Asia. The soil where these trees grow influences the appearance of some macro and micro-elements in the chemical structure of the Dammar resin, meaning that additional elements may appear too, such as: iron, silicon, zirconium, cadmium, etc. Dammar films, in the form of varnish, are used for painting protection (see [33]). The effect of long exposure to acetic acid of the painting covered with Dammar-based varnish, which generates their accelerated damage, has been examined in [34,35].

In paper [36] the nature of the macromolecular constituents of the Dammar resin was clarified using a multi-analytical approach based on thermo-gravimetric analysis and infrared spectroscopy (TGA-FTIR).

Dammar resin is known to be part of certain drugs. Thus, ref. [37] looks into the continuous administration of the drug atenolol, made of Dammar gum interlaced with hydrogel biodegradable composites based on polyacrylamide and zirconium. The interlacing process was synthesized successfully, obtaining improved properties for the polyacrylamide Dammar gum, which is explained by the presence of an inorganic component in the zirconium mixture. The hydrogel properties (Dammar gum combined with zirconium iodoxalate) are analyzed in [38,39,40]. Detailed studies on its structure and chemical composition are conducted in [41,42].

Dammar has also been used in order to obtain certain ecological binders, made of silicon and dammar, with improved properties, which are shown in [43]. There, the optimal composition of these types of binders was determined, ensuring the best properties for impact stress, hardness, traction and adherence. The way in which Dammar addition contributed to improving the rigidity, the modulus of elasticity and the hardness of modified silicon analyzes was investigated in paper [44].

Dammar-based hybrid resins have been used recently to obtain composite materials reinforced by natural fibers. In [28] the hybrid resins obtained by combining Dammar, epoxy resin and composite materials made of these resins with hemp, cotton, flax, straw and cattail reinforcements were studied. The mentioned work analyzes the influence of the epoxy resin volume ratio on the mechanical behavior of the bio resins and composite materials under study. Some mechanical properties and the non-uniformity impact on the mechanical behavior of composites with Dammar-based hybrid resin matrix, reinforced by hemp fabric were analyzed in [45].

Particular emphasis was put on material re-use and recycling. Thus, this work analyzes the properties of certain hybrid resins with a Dammar volume ratio of 60%, 70% and 80% respectively. In addition, the mechanical behavior of composite materials with a matrix made of the three types of the above-mentioned resins were studied, and hardener is made of waste paper sheets and shredded waste paper respectively.

The third purpose of this article is to find new possibilities for the use of paper waste and to determine the limits of applicability of composites with this type of reinforcer. Based on the determined properties, we will propose using these composite materials in various fields of activity.

## 2. Materials and Methods

### 2.1. Specimens Manufacturing

The temperature of the casting environment was 21–23 ${}^{\circ }$C. All samples were cut out 10 days after casting the plates. The first step consisted of casting hybrid resin plates, where we used a Dammar volume ratio of 60%, 70% and 80%, respectively. The difference up to 100% consisted of epoxy resin of the Resoltech 1050 type, together with its associated reinforcement of the Resoltech 1055 type. The synthetic component (epoxy resin and hardener) were necessary in order to generate a quick process of polymerization.

The volume ratio of the mixture between Resoltech 1050/Resoltech 1055 was 7/3 (according to the manufacturer’s instructions). The thermo-mechanical properties of the epoxy resin Resoltech 1050, together with the hardener Resoltech 1055, are given by the manufacturer (see [46]).

From each cast plate, a set of ten samples were cut out. These samples were labelled Da 1.1–10, Da 2.1–10, Da 3.1–10 (see Figure 1). The size of the samples was: 250 mm long, 25 mm wide, 6.2 mm thick for the hybrid resin samples. The sample density ranged between 1.04–1.05 g/cm${}^{3}$.

The second step consisted in using epoxy resin and the three types of hybrid resin and casting four plates of composite materials reinforced by 18 layers of waste paper sheets (weighing 80 g in total). The thickness of a sheet of paper was 0.0105 mm and the specific mass was 80 g/m${}^{2}$. The 18 layers of paper were pre-impregnated with resin. They were placed successively, one on top of the other, and at the end a uniform pressure of 27,000 N/m${}^{2}$ was applied on the laminate. A set of ten samples, labelled 0.0.1–10, 0.1.1–10, 0.2.1–10, 0.3.1–10, of each plate were cut out (see Figure 2). The sample sizes were: 250 mm long, 25 mm wide and 2.6 mm thick. The sample density ranged between 1.12–1.18 g/cm${}^{3}$, and the resin mass ratio between 0.49 and 0.52.

The third step consisted of using epoxy resin and the three types of hybrid resin and casting four plates of composite materials reinforced by shredded waste paper. A document shredder was used to obtain the shredded paper, which cut the sheets of paper into 4 mm wide strips. Then, the obtained strips were randomly cut into pieces with a length between 10–20 mm. A variable speed Steinhaus PRO-MX850 paint/mortar mixer was used to obtain a homogeneous mixture (resin-paper). A uniform pressure of 27,000 N/m${}^{2}$ was applied to the plates.

The four sets of ten samples each, labelled A.0.1–10, A.1.1–10, A.2.1–10, A.3.1–10 (see Figure 3) were cut out. The sample sizes were 250 mm long, 25 mm wide and 5.2 mm thick. The sample density ranged between 1.09–1.13 g/cm${}^{3}$, and the resin mass ratio between 0.58 and 0.60.

These composites required a larger ratio of resin so as to obtain a better homogenization of the mixture.

### 2.2. Methods of Analysis and Equipment

All samples made in Section 2.1 were submitted to the tensile test, Scanning Electron Microscopy (SEM) analysis and vibration analysis. Moreover, the hybrid resin samples were submitted to the FTIR and EDS analysis.

#### 2.2.1. Tensile Test

The tensile test met the requirements of the ASTM D3039 standard (see [47]). The Lrx Plus testing machine from LLOYD Instruments with a force range of 2.5 kN, and the NEXYGEN analysis software (see [48] with the manufacturer’s technical specifications) were used for this trial.

The elements obtained from this test were: tensile response, tensile strength $R_{m}$ [MPa], percentage elongation at break A[%] and modulus of elasticity *E* [N/mm${}^{2}$].

#### 2.2.2. FTIR, EDS and SEM Analysis

The functional groups and the chemical composition of the three types of hybrid resin were determined by the FTIR analysis the EDS (Energy, Dispersive X-ray Spectrometry) analysis respectively.

The FTIR analysis was carried out with the help of a portable IdentifyIR system (fitted with the ChemAssist software). The requirements of the ASTM E168 and ASTM E1252 standards have been met (see [49,50]). The sample size was under 5 μL or 100 μm. In addition, the equipment was fitted with the ATR spectral libraries for Aldrich/Smiths detection. All technical specifications of this system can be found on the manufacturer’s webpage (see [51]).

The microscopic study and the study of the structure phases, both for hybrid resin samples and the analyzed composite material samples were conducted on the basis of the SEM analysis. The requirements of the ASTM F2603 standard have been met (see [52]).

The SEM-EDS analysis was carried out with a Hitachi S3400N-tip N electronic microscope, fitted with EDX Oxford Instruments X-act, with conventional cathode (see [53] with the manufacturer’s technical specifications). The requirements of the ASTM 1508 standard have been met for the EDS analysis (see [54]).

#### 2.2.3. The Vibration Analysis

The damping coefficient and the natural frequency for all samples were experimentally measured. The samples were clinched at one end and the measurement was done at the free end. The free length of the samples was 120, 140, 160, 180, 200 and 220 mm.

The following equipment was used for the vibration measurement: the SPIDER 8 data acquisition system, connected via a USB port to notebook; data acquisition was made with the help of a CATMAN EASY software, which connected two units; a NEXUS 2692-A-0I4 conditioning amplifier, connected to SPIDER 8; a 0.04 pC/ms${}^{-2}$ accelerometer, connected to the conditioning amplifier.

The frequency range was set from 0 to 2.400 Hz in SPIDER 8. At the beginning of the experimental determinations, “Automatic zero balance” was set to bring the accelerometer signal to the point of “perfect zero”. A performed a Butterworth “High Pass” filtering of each measurement, at a 3 Hz frequency was performed to exclude the errors brought up by the experimental system.

## 3. Results

Henceforward, the sample with the average values of the studied mechanical properties was defined by representative sample.

### 3.1. The Experimental Results for the Hybrid Resins Studied

In this section, the experimental results for the three types of hybrid resin samples (whose fabrication method was described in Section 2.1) were presented, which were submitted to the tensile test, FTIR, EDS and SEM analyzes and the vibration analysis.

Figure 4 shows the tensile response of a representative sample from the set of hybrid resin samples with a Dammar volume ratio of 60% (set Da 1.x); 70% (set Da 2.x); 80% (set Da 3.x).

Table 1 shows the mean value with the standard deviation for the modulus of elasticity *E* [N/mm${}^{2}$], tensile strength $R_{m}$ [MPa] and elongation at break A[%], for the sample sets of the types Da 1.x, Da 2.x and Da 3.x.

We noticed an important modification of mechanical behavior when the ratio of the natural resin Dammar was changed. The modulus of elasticity with 60% Dammar was double, compared to the modulus of elasticity with 80% Dammar hybrid resin. The modification was even higher in the case of tensile strength. Thus, the tensile strength for the 60% Dammar hybrid resin was almost three times higher than the tensile strength of the 80% Dammar hybrid resin. Moreover, the tensile response also changed. If, in the case of 60% Dammar hybrid resin, the tensile response was almost linear along its whole length, as in the case of the 70% and 80% Dammar hybrid resins, nonlinearities appeared when the stress was increased and the resins had a plastic behavior. Based on the results reported in Table 1, the addition of Dammar resin makes the resin mixture more ductile while lowering the tensile strength and modulus of elasticity. This phenomenon can be explained by the fact that the resins, even in solid-state, have a rheological behavior characterized by viscosity. Due to external stresses, a low viscosity leads to significant deformations. Dammar resin diluted with turpentine remains liquid, so it has a low viscosity. Solidification occurs by mixing with synthetic resin. After curing, the resulting mixture will have a rheological behavior whose viscosity decreases with the increasing volume ratio of Dammar. Therefore, at the same external load, the deformations of the hybrid resin by 80% Dammar were higher than the deformations of the hybrid resin by 70%, respectively 60% volume ratio of Dammar.

In-depth studies on the composition and chemical properties of the hybrid resin-based on Dammar (with various volume proportions) were performed in the paper [55].

The hybrid resin with a volume ratio of 70% Dammar (from the point of view of the chemical structure) was studied using the FTIR analysis, because it fell within the average of the chemical properties of the three types of hybrid resin.

Figure 5 shows the FTIR analysis of a hybrid resin specimen (70% Dammar volume ratio) and of the same specimen re-tested. The characteristic bands and a list of peaks observed in the interval 4000–650 cm${}^{-1}$ are shown in infrared.

Based on the diagram (and the peak list) two functional groups stand out:Ketene type (CH${}_{2}$=C=O) cumulated double bonds, which are substances with cumulated double bonds of carbonyl and carbon–carbon;double bond of nitrogen, known as the nitrite ion (NO${}_{2}^{-}$), which contains nitrogen in a state of relatively unstable oxidation.

Table 2 shows 10 volatile compounds identified in the chemical structure of hybrid resin (with a 70% Dammar volume ratio).

Based on the EDS analysis of the hybrid resin specimen (with a 70% Dammar volume ratio), extracted from type Da 2.x samples, Figure 6 shows the diagram of the chemical composition obtained at 5 keV intensity.

On the basis of the EDS analysis, Table 3 shows the chemical composition of a hybrid resin specimen (with a 70% Dammar volume ratio), taken from the Da 2.x set. This composition is expressed by the weight concentration, atomic concentration and atoms number in each element.

A total number of atoms approximately equal to $48\cdot 10^{23}$ was used.

Just as in [28], after analyzing the chemical concentration of the three types of resin (with Dammar volume ratio of 60%, 70% and 80% respectively), the conclusions are that:-in Carbon there is a decrease in the Atomic Concentration and the Weight Concentration as Dammar volume ratio increases;-in Oxygen there is an increase in the Atomic Concentration and the Weight Concentration as the Dammar volume ratio increases.

These conclusions are also backed up by the results of the SEM analysis for the three types of hybrid resin and the epoxy resin presented in Figure 7.

Based on the SEM analysis of the three types of hybrid resin can be seen that the number of voids is higher when the Dammar volume ratio is higher. There are several factors influencing the presence of the voids in the cured laminate. One is the viscosity of the resin after the Dammar is introduced. After mixing, the Dammar increased the viscosity of the resin system, then, although the cure reaction takes much longer, the higher viscosity slows down the removal of the voids. Another possibility is that voids form during the polymerization process. In the case of epoxy resin, the hardener produced a rapid reaction, the voids being removed in a short time, before the complete hardening of the resin (which occurred in less than 24 h). When the epoxy resin was mixed with Dammar natural resin, the hardening time was much longer (72–96 h). In this way, due to the high viscosity, the voids that appeared during the reaction could no longer be removed and remained trapped in the resin.

### 3.2. The Experimental Results for the Composite Materials

In this section, the experimental results obtained for the composite material samples whose fabrication method was described in Section 2.1 (those with a matrix made of 70% Dammar) and which were submitted to a tensile test and vibration analysis were presented.

Figure 8 shows the tensile response for a representative sample of composite materials reinforced by 18 layers of waste paper, epoxy resin and, respectively, a hybrid resin matrix with a Dammar volume ratio of 60%, 70% and 80%.

Figure 9 shows the tensile response for a representative sample of composite materials reinforced by shredded waste paper, with an epoxy resin and, respectively, hybrid resin matrix with a Dammar volume ratio of 60%, 70% and 80%.

Table 4 shows the mean value with the standard deviation for the modulus of elasticity *E* [N/mm${}^{2}$], tensile strength $R_{m}$ [MPa] and elongation at break *A* [%], for the sets of composite material samples.

An obviously layered distribution for the composites made of waste paper sheets and a random distribution for the composites made of shredded waste paper were noticed. This difference in the reinforcing material distribution leads to important differences in mechanical behavior too.

Figure 10 shows the image of the break area for a representative sample of the composite material set 0.0.x (a), 0.1.x (b), 0.2.x (c), 0.3.x (d).

Figure 11 shows the image of the break area for a representative sample of the composite material set A.0.x (a), A.1.x (b), A.2.x (c), A.3.x (d).

### 3.3. The Vibration Behavior

The presence of some mechanisms of energy dissipation is accepted now in all models used for simulating mechanical vibrations in elastic systems. There are two types of damping: external (due to the interaction with the environment, or with other physical systems) and internal (caused by processes within the system, such as an increase in thermal energy to the detriment of mechanical energy, which is due to internal friction). The influence of the mechanisms of energy dissipation is taken into account by inserting additional terms in the movement equations. A comparative analysis referring to the measurement methods of energy dissipation for plates with uniform layers and viscoelastic damping was investigated in [56]. The damped bending vibrations for composite plates of glass fiber containing one or several holes were studied in [57]. The damping properties of laminated structures made of steel, rubber and epoxy resin reinforced by glass fiber have been studied in [58,59]. The individual loss factors of the component materials have been used so as to determine the loss factors of hybrid structures. Other studies on the damped vibrations of composite bars can be found in [60,61,62].

The logarithmic decrement method was used in order to measure the system damping.

The experimental recording of the free vibrations in a certain point allows calculating the damping factor μ, with the help of the formula (see [63]):
(1)$$\mu =\frac{1}{\Delta t}\ln\frac{w_{1}}{w_{2}},$$
where:-$w_{1}$ and $w_{2}$ are the maximum values in the diagram, chosen for calculating the damping factor;-Δt is the time interval between the two maximum values.

Since the plate damping in free damped vibrations is a combination of structural damping and damping due to air friction in all the recordings, to determine the damping factor μ, the areas where the vibration damping ranged between the amplitudes of $w_{1}=2$ mm land $w_{2}=0.05$ mm were selected.

To measure vibrations, each bar was embedded at one end and the measuring transducer was mounted at the free end. The bar was deformed by applying a force to the free end. The free vibrations obtained by eliminating the force specified above were measured.

During the free vibrations that have been measured, the stresses that occurred were small and therefore the materials from which the bars were made were required in the elastic domain, where there is proportionality between specific stresses and strains (in this area Hooke’s law is valid). Figure 12 shows the vibration recording (natural frequency and damping factor) for a sample of the Da 2.x set, with a free length of 120 mm.

At the recording in Figure 12, for the determination of the damping factor, the first marker (red dotted line in the graph) was placed on the maximum with the value $w_{1}=1.9111$, corresponding to the time moment $t_{1}=1.5413$ and the second marker (blue dotted line in the graph) on the maximum with the value $w_{2}=1.1046$, corresponding to the time moment $t_{2}=1.6150$. The damping factor was calculated with the Formula (Equation 1). The damping factor was similarly calculated for all experimental measurements.

Table 5 shows the vibration behavior of the epoxy resin samples and the three types of Dammar samples with a length of 25 mm. The shown values represent the arithmetic mean for three measurements.

Figure 13 shows the vibration recording (natural frequency and damping factor) for a sample of the 0.2.x set, with a length of 120 mm.

Figure 14 shows the vibration recording (natural frequency and damping factor) for a sample of the A.2.x set, with a length of 120 mm.

Table 6 shows the vibration behavior of the composite material samples reinforced by layers of waste paper sheets. The values represent the arithmetic mean for three measurements.

Table 7 shows the vibration behavior of the composite material samples reinforced by shredded waste paper. The values represent the arithmetic mean for three measurements.

The natural frequencies of the bars depend on not only the dimensions (thickness, length), but also on the material properties (density and modulus of elasticity). Because the test samples studied had similar dimensions, the differences between the measured frequencies are due to the differences between the modulus of elasticity of the composite materials from which the samples are made. There is a correspondence between the measured frequencies and the modulus of elasticity determined at the tensile test, more precisely the increase of the modulus of elasticity leads to the increase of the vibration frequency. The damping factor as a whole characterizes the vibration damping capacity for a test piece. The vibration damping capacity increases with an increasing volume ratio of Dammar in the hybrid resin used as a matrix.

## 4. Discussion and Conclusions

Using natural resins to make composite materials is influenced by the properties of these resins and their capacity for creating a synergetic effect together with the reinforcing materials. The analysis of the results shows an important variation of the hybrid resin properties depending on the ratio between natural and synthetic resins. Thus, the value of the modulus of elasticity decreases quickly as the Dammar ratio in the hybrid resin composition increases, from 1720 (±90) MPa for hybrid resin with 60% Dammar to 835 (±75) MPa for hybrid resin with 80% Dammar. The value of the tensile strength decreases from 20.2 (±0.7) MPa for the hybrid resin with 60% Dammar to 7.8 (±0.7) MPa for the hybrid resin with 80% Dammar. The value of the elongation at break increases as the Dammar ratio rises up.

There are significant changes in the tensile response too. If, in the case of hybrid resin with 60% Dammar, the tensile response has an almost linear character (nonlinearities appear in the final part only), in the hybrid resin with 80% Dammar the nonlinearity is significant from the very beginning of the stress.

The following structural identification are obtained based on the FTIR analysis of a hybrid resin spectrum (with a 70% Dammar volume ratio):-3000–2800 cm${}^{-1}$ a stretch of the aromatic and aliphatic C-H bonds occurs;-1608 cm${}^{-1}$ a stretch of the C-C bonds of aromatic rings occurs;-1507 cm${}^{-1}$ a stretch of the aromatic C-C bonds occurs;-1449 cm${}^{-1}$ indicates the presence of methylene groups (CH${}_{2}$);-the additional presence of the peak at 1375 cm${}^{-1}$ is caused by a methyl group (CH${}_{3}$);-1034 cm${}^{-1}$ a stretch of the C-O-C ethers occurs;-915 cm${}^{-1}$ a stretch of the oxiranic C-O group occurs;-831 cm${}^{-1}$ a stretch of the oxiranic C-O-C groups occurs;-772 cm${}^{-1}$ a rocking movement of the CH${}_{2}$ occurs.

A conclusion of the FTIR analysis of the spectra of the three types of resin was noticed refers to that there is only mitigation of the absorber and there are no extinctions or displacements of the strips, which suggests a lack of interaction between components.

As far as the studied composite materials are concerned the comparison of the experimental results shows an important change in the mechanical properties too when the Dammar volume ratio changes. A decrease in the values for tensile strength and modulus of elasticity with increasing of Dammar volume ratio of natural resin in the mixture is observed However, there are significant differences in the mechanical behavior between the composite materials reinforced by layers of waste paper sheets and the composite materials reinforced by randomly-laid shredded waste paper. Thus, for composite materials reinforced by layers of waste paper sheets, the tensile strength is much higher than the tensile strength of the resin in the case in which was used as a matrix. This shows that the charge is taken over by the paper layers on which the stress is uniformly applied.

Instead, for the composite materials with shredded waste paper, the tensile strength is a little higher than the resin tensile strength. This phenomenon is explained as such: the random layout of the paper reinforcement creates some sections of the samples where the stress is taken over to a little degree by the reinforcing material. This can also explain, in the case of composite materials reinforced by shredded waste paper, that the tensile strength has a much lesser increase than the modulus of elasticity, compared to the samples of the three types of hybrid resin. The phenomenon is due to the non-uniformities appearing in the reinforcing material distribution and is explained in more detail in [45].

Important differences between the two types of composite materials also occur in the shape of the tensile response. Thus, the tensile response is practically linear throughout the entire stress in the case of the composite materials reinforced by shredded waste paper. This shows that during the trial, the reinforcing material and the matrix take over the stress together. The fact that the elongations at the breaks of these composites are lower than the elongations at the breaks of the hybrid resins used as matrices shows that, due to non-uniformities, there are areas where the stress is mainly taken over by matrix and the elongation at break of the hybrid resin is overcome. In the case of the composite materials reinforced by layers of waste paper sheets, only the first part of the tensile response is linear whereas important nonlinearities occur for specific elongations bigger than 0.5%. There are several explanations for this phenomenon. The first is the nonlinear behavior of the reinforcer made of waste paper sheets. Then, when the natural deformation increases, the hybrid resins used as a matrix display plastic deformations. Moreover, when the stress tensions increase, these are mainly taken over by the paper sheets, which have a bigger tensile strength than the hybrid resins.

As to the vibration behavior, an increase in the damping capacity is observed when the Dammar ratio in the composition increases.

Since the damping factor depends on the sample length, it characterizes the entire damping capacity of the sample. The damping factor is inversely rational with the square of the free length of the bar. Since there is a similar dependence in the case of the own pulsations, our conclusion was that it is predominantly a mechanism of energy dissipation, where the damping force is rational to the bending speed of the bar. In order to ascertain the damping capacity of the resins and composite materials under study, the loss factor $\eta =\frac{\mu }{\pi \nu }$ can be calculated (see [28]). The following average results are obtained:-for 60% Dammar resin η=0.0247;-for 70% Dammar resin η=0.0329;-for 80% Dammar resin η=0.0447;-for the composite materials reinforced by waste paper sheets and a matrix made of epoxy resin η=0.0227;-for the composite materials reinforced by shredded waste paper sheets and a matrix made of hybrid resin with 60% Dammar η=0.0254;-for the composite materials reinforced by waste paper sheets and a matrix made of hybrid resin with 70% Dammar η=0.0397;-for the composite materials reinforced by waste paper sheets and a matrix made of hybrid resin with 80% Dammar η=0.0431;-for the composite materials reinforced by shredded waste paper and a matrix made of epoxy resin η=0.0267;-for the composite materials reinforced by shredded waste paper and a matrix made of hybrid resin with 60% Dammar η=0.0353;-for the composite materials reinforced by shredded waste paper and a matrix made of hybrid resin with 70% Dammar η=0.0428;-for the composite materials reinforced by shredded waste paper and a matrix made of hybrid resin with 80% Dammar η=0.0503.

The loss factor almost doubles in the composite materials with hybrid resin and 80% Dammar, compared to the composite resins in the epoxy resin. The properties of the materials reinforced by shredded waste paper and a matrix made of hybrid resin recommend their use in the medical field, for example, to make re-usable medical devices that can be helpful for the immobilization, limitation, or assistance regarding certain areas of the bone system, affected by fractures, sprains, so on and so forth. These composite materials are surely better accepted by the human body because they consist mainly of natural elements.

The properties of the composite materials reinforced by continuous layers of waste paper, with a matrix made of hybrid resin, are recommended to be employed, for example, in the car industry in order to make the interior lining of car doors, or certain dashboard components, so on and so forth.

## Figures and Tables

**Figure 1 polymers-12-01688-f001:**
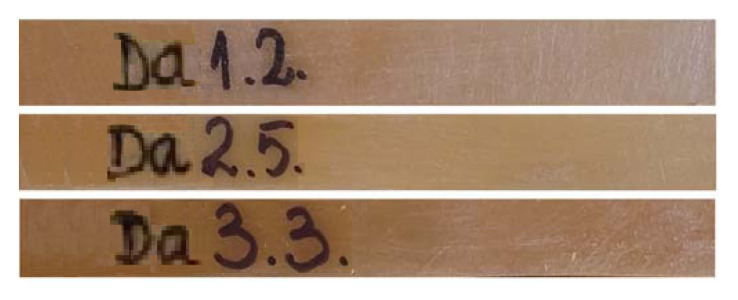
Hybrid resin sample with a volume ratio of 60% (Da 1.2), 70% (Da 2.5), and 80% Dammar (Da 3.3) respectively.

**Figure 2 polymers-12-01688-f002:**
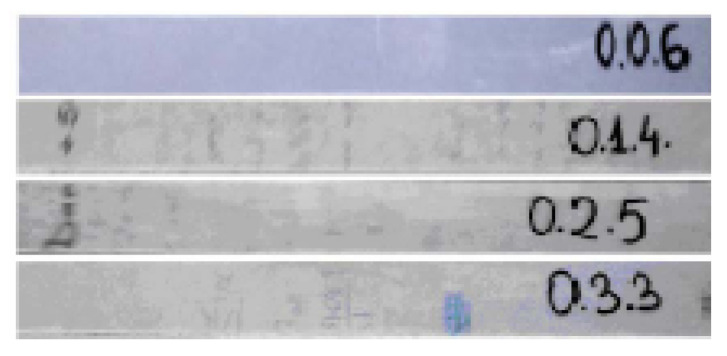
Samples of composite materials reinforced by waste paper sheets, with an epoxy resin matrix (0.0.6) and, respectively, hybrid resin with a Dammar volume ratio of 60% (0.1.4), 70% (0.2.5) and 80% (0.3.3).

**Figure 3 polymers-12-01688-f003:**
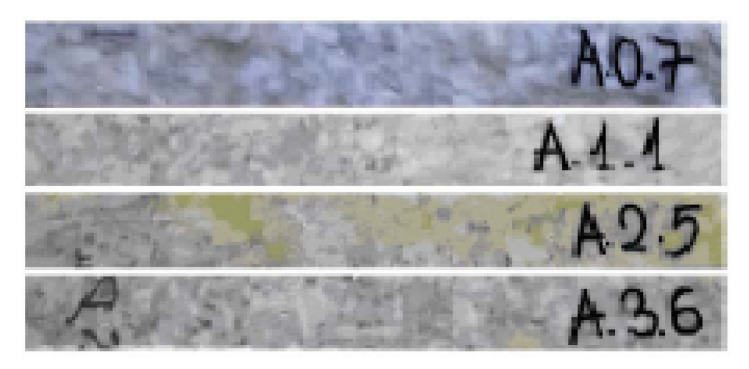
Samples of composite materials reinforced by shredded waste paper, with an epoxy resin matrix (A.0.7) and, respectively hybrid resin with a Dammar volume ratio of 60% (A.1.1), 70% (A.2.5) and 80% (A.3.6).

**Figure 4 polymers-12-01688-f004:**
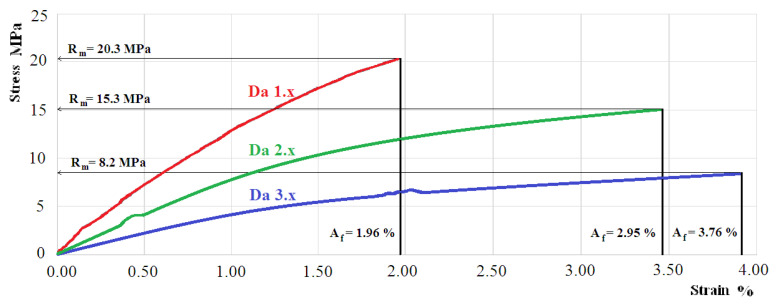
Tensile response for a representative sample of the set Da 1.x; Da 2.x; Da 3.x.

**Figure 5 polymers-12-01688-f005:**
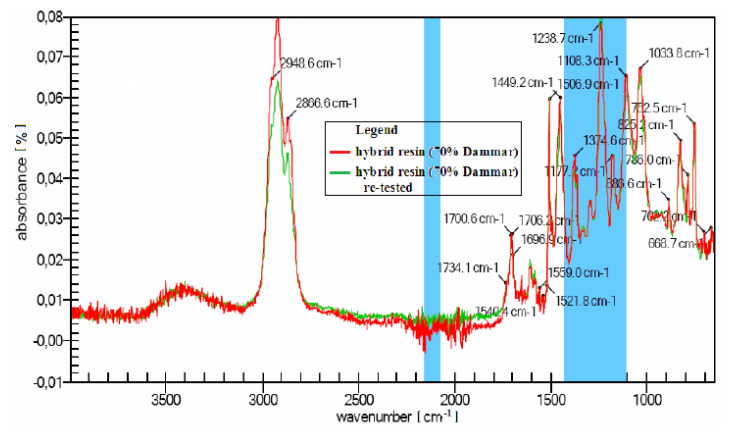
The FTIR analysis of a hybrid resin specimen (70% Dammar) and of the same specimen re-tested.

**Figure 6 polymers-12-01688-f006:**
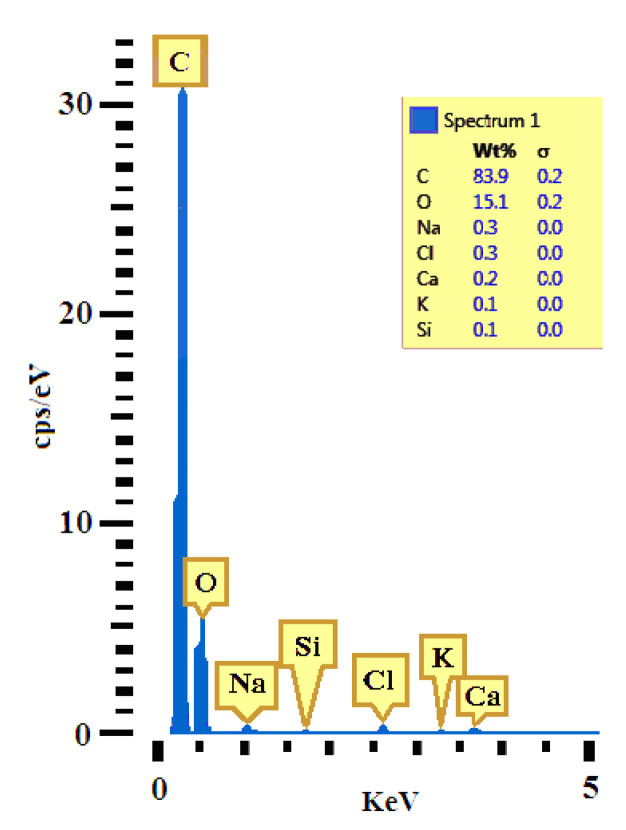
Diagram of the Energy, Dispersive X-ray Spectrometry (EDS) analysis of the chemical composition of a hybrid resin specimen (70% Dammar), taken from the Da 2.x set, obtained at 5 keV intensity.

**Figure 7 polymers-12-01688-f007:**
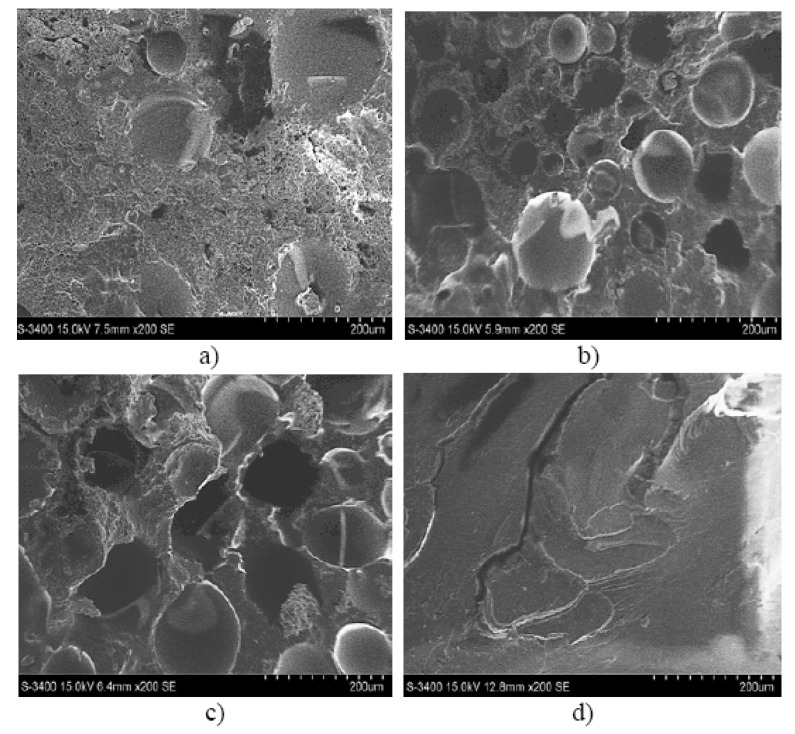
Scanning Electron Microscopy (SEM) analysis for a sample type: (**a**) Da 1.x, (**b**) Da 2.x, (**c**) Da 3.x (**d**) epoxy resin.

**Figure 8 polymers-12-01688-f008:**
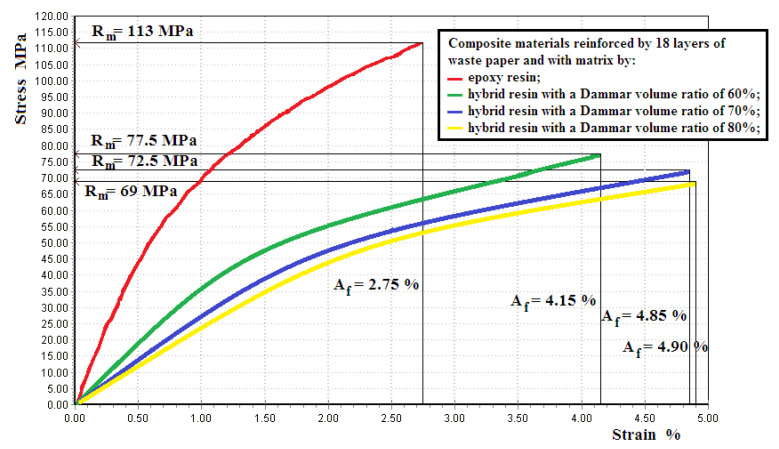
The tensile response for a representative sample of composite materials reinforced by 18 layers of waste paper and with an epoxy resin and, respectively, hybrid resin matrix with a Dammar volume ratio of 60%, 70% and 80%.

**Figure 9 polymers-12-01688-f009:**
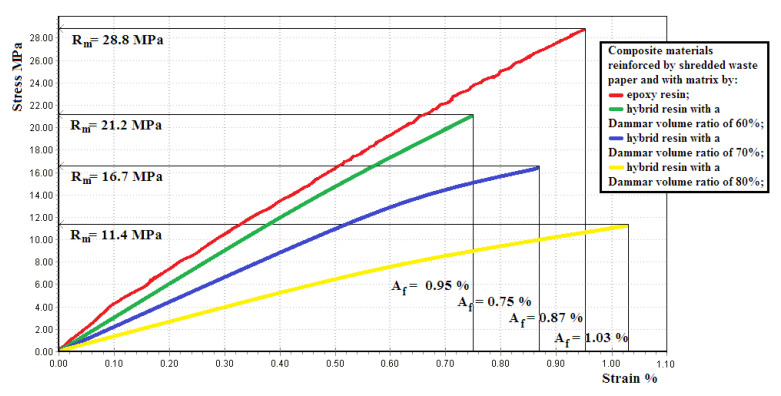
The tensile response for a representative sample of composite materials reinforced by shredded waste paper and with an epoxy resin and, respectively, hybrid resin matrix with a Dammar volume ratio of 60%, 70% and 80%.

**Figure 10 polymers-12-01688-f010:**
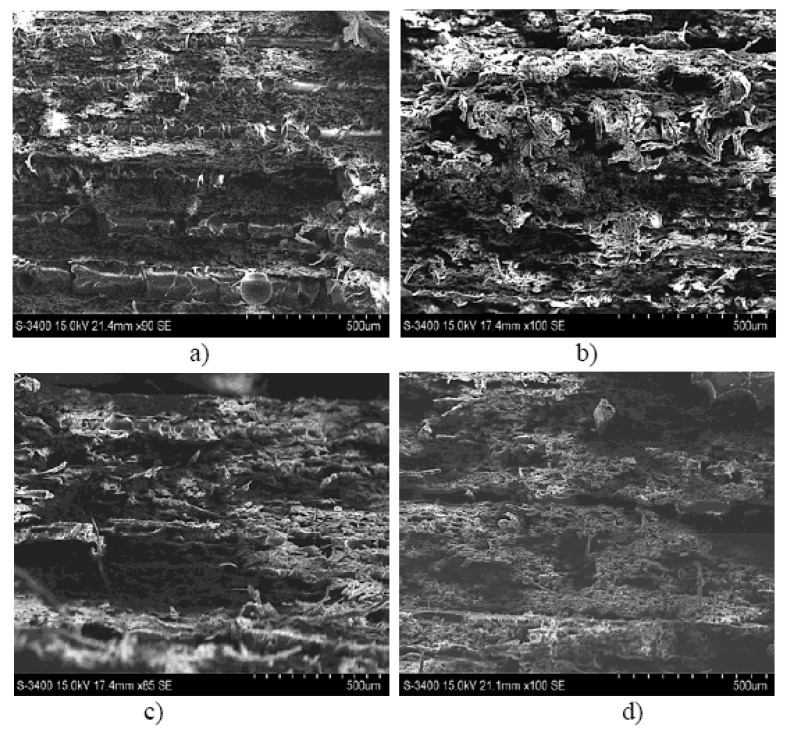
The image of the break area for a representative sample of the composite material set 0.0.x (**a**), 0.1.x (**b**), 0.2.x (**c**), 0.3.x (**d**).

**Figure 11 polymers-12-01688-f011:**
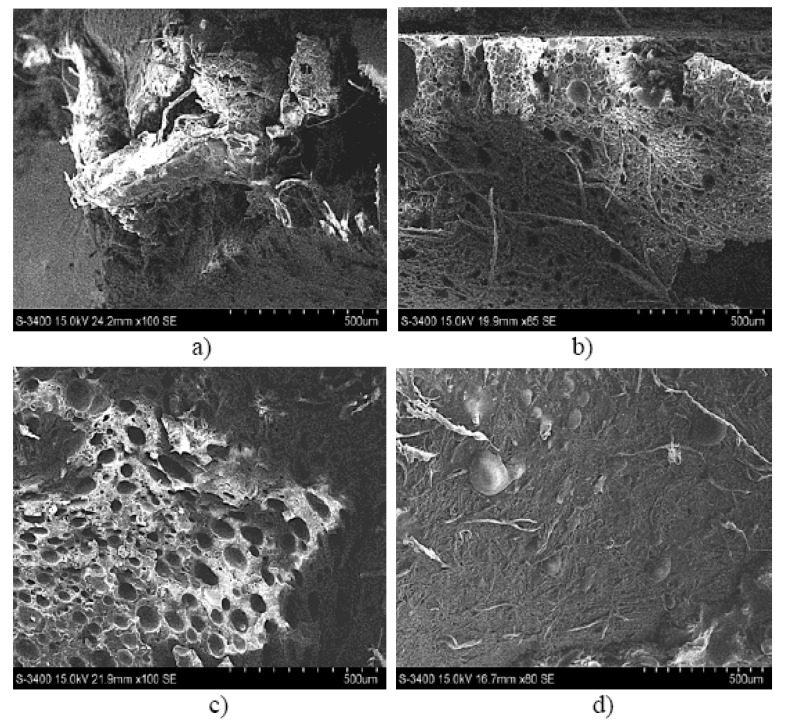
The image of the break area for a representative sample of the composite material set A.0.x (**a**), A.1.x (**b**), A.2.x (**c**), A.3.x (**d**).

**Figure 12 polymers-12-01688-f012:**
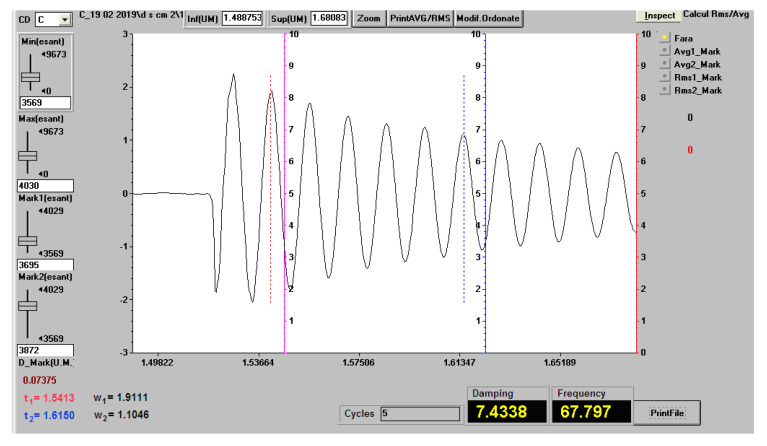
The vibration recording (natural frequency and damping factor) for a sample of the Da 2.x set, with a free length of 120 mm.

**Figure 13 polymers-12-01688-f013:**
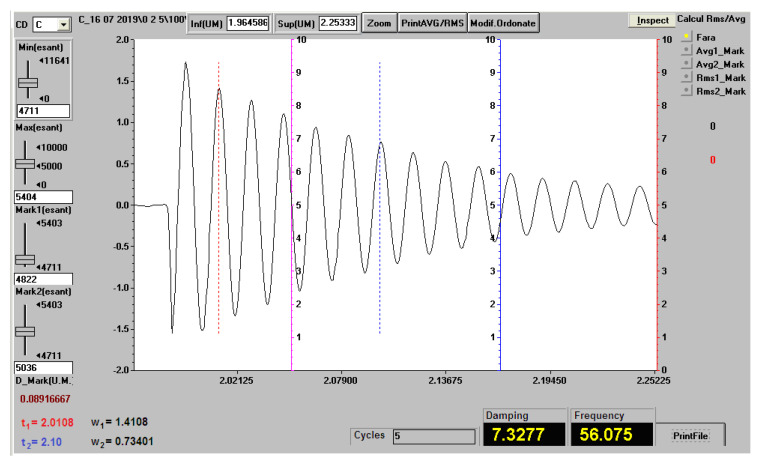
Vibration recording (natural frequency and damping factor) for a sample of the 0.2.x set, with a length of 120 mm.

**Figure 14 polymers-12-01688-f014:**
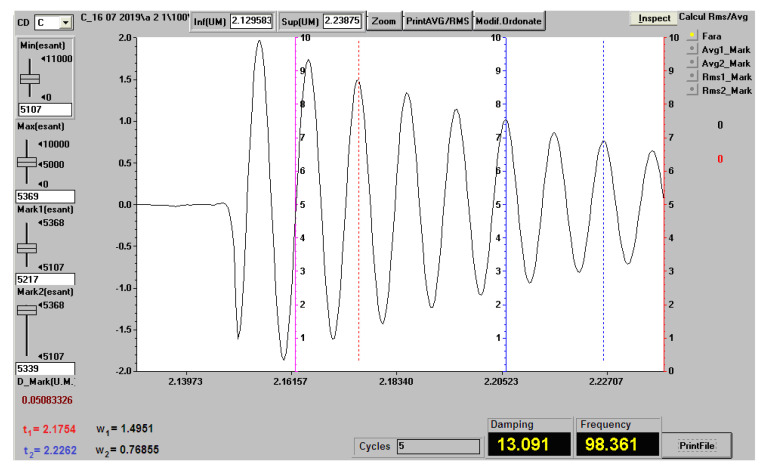
Vibration recording (natural frequency and damping factor) for a sample of the A.2.x set, with a length of 120 mm.

**Table 1 polymers-12-01688-t001:** The mean value with the standard deviation for the modulus of elasticity, tensile strength and elongation at the break for the hybrid resin samples.

Sample	Density	Modulus of Elasticity	Tensile Strength	Elongation at Break
Type	ρ [g/cm${}^{3}$]	*E* [N/mm${}^{2}$]	$R_{m}$ [MPa]	*A* [%]
Da 1.x	1.06	1720 (±90)	20.2 (±0.7)	1.97 (±0.08)
Da 2.x	1.05	1260 (±70)	15.0 (±0.8)	2.95 (±0.08)
Da 3.x	1.04	835 (±70)	7.2 (±0.7)	3.80 (±0.11)

**Table 2 polymers-12-01688-t002:** The 10 volatile compounds identified in the chemical structure of hybrid resin (with a 70% Dammar volume ratio).

Current	Spectrum	Aldrich/Smiths Detection ATR
Number		Special Libraries
1	0.7929	Geranium algerie; Essence
2	0.7757	Geranium bourbon (Natural Essential Oil)
3	0.7732	Vetiver (Natural Essential Oil)
4	0.7707	Dammar Gum
5	0.7693	Vetiver java (Natural Essential Oil)
6	0.7528	Peppermint oil
7	0.7487	Patchouli (Natural Essential Oil)
8	0.7177	Magna lemon oil (Natural Essential Oil)
9	0.7165	Dinonylnaphthalenesulfonic acid (in kerosene)
10	0.6974	Romarin (Natural Essential Oil)

**Table 3 polymers-12-01688-t003:** The chemical composition of a hybrid resin specimen (with a 70% Dammar volume ratio), taken from the Da 2.x set.

Element	Element	Element	Atomic	Weight	Atomic
Number	Symbol	Name	Weight	Conc.	Conc.
				[%]	[%]
6	C	Carbon	12	83.9	87.72
8	O	Oxygen	16	15.1	11.84
11	Na	Sodium	23	0.3	0.16
17	Cl	Chlorine	35.5	0.3	0.12
20	Ca	Calcium	40	0.2	0.065
19	K	Potassium	39	0.1	0.045
14	Si	Silicon	28	0.1	0.05

**Table 4 polymers-12-01688-t004:** The mean value with the standard deviation for the modulus of elasticity, tensile strength and elongation at break for the sets of composite material samples.

Sample	Resin Mass	Density	Modulus of Elasticity	Tensile Strength	Elongation at Break
Type	Ratio	ρ [g/cm${}^{3}$]	*E* [N/mm${}^{2}$]	$R_{m}$ [MPa]	*A* [%]
0.0.x	0.52	1.18	5345 (±125)	111.8 (±2.2)	2.77 (±0.09)
0.1.x	0.51	1.14	4340 (±90)	76.4 (±1.7)	4.33 (±0.19)
0.2.x	0.51	1.14	3995 (±115)	71.3 (±1.5)	4.77 (±0.14)
0.3.x	0.49	1.12	3490 (±90)	68.9 (±1.6)	4.93 (±0.12)
A.0.x	0.60	1.13	3250 (±100)	28.0 (±1.1)	0.97 (±0.06)
A.1.x	0.58	1.10	2475 (±125)	20.8 (±0.9)	0.76 (±0.05)
A.2.x	0.58	1.09	2025 (±75)	16.4 (±0.7)	0.87 (±0.05)
A.3.x	0.59	1.09	1290 (±80)	11.0 (±0.6)	1.03 (±0.05)

**Table 5 polymers-12-01688-t005:** The vibration behavior of the hybrid resin samples.

Free	Da 1.x (60% Dammar)		Da 2.x (70% Dammar)		Da 3.x (80% Dammar)	
Length	Frequency	Damping	Frequency	Damping	Frequency	Damping
[mm]	ν [Hz]	μ [s${}^{-1}$]	ν [Hz]	μ [s${}^{-1}$]	ν [Hz]	μ [s${}^{-1}$]
120	77.4	6.12	67.8	7.40	56.5	7.96
140	56.9	4.42	49.7	5.61	41.9	6.06
160	43.2	3.23	38.9	4.12	32.0	4.47
180	34.6	2.68	29.7	3.15	25.4	3.48
200	27.7	2.11	24.6	2.17	20.6	2.93
220	23.1	1.85	20.2	1.87	17.0	2.35

**Table 6 polymers-12-01688-t006:** The vibration behavior of the composite material samples reinforced by layers of waste paper sheets.

Free	0.0.x (Epoxy Resin)		0.1.x (60% Dammar)		0.2.x (70% Dammar)		0.3.x (80% Dammar)	
Length	Frequency	Damping	Frequency	Damping	Frequency	Damping	Frequency	Damping
[mm]	ν [Hz]	μ [s${}^{-1}$]	ν [Hz]	μ [s${}^{-1}$]	ν [Hz]	μ [s${}^{-1}$]	ν [Hz]	μ [s${}^{-1}$]
120	64.8	4.75	61.2	4.91	55.8	7.29	51.4	7.39
140	47.3	3.42	45.4	3.65	40.6	5.24	36.9	5.25
160	36.1	2.57	34.9	2.89	34.8	3.98	28.5	3.98
180	28.5	2.01	27.6	2.23	24.3	3.02	23.0	2.95
200	23.2	1.64	22.8	1.77	20.2	2.48	18.9	2.47
220	19.2	1.35	19.1	1.45	16.3	2.06	15.4	1.98

**Table 7 polymers-12-01688-t007:** The vibration behavior of the composite material samples, reinforced by layers of shredded waste paper.

Free	A.0.x (Epoxy Resin)		A.1.x (60% Dammar)		A.2.x (70% Dammar)		A.3.x (80% Dammar)	
Length	Frequency	Damping	Frequency	Damping	Frequency	Damping	Frequency	Damping
[mm]	ν [Hz]	μ [s${}^{-1}$]	ν [Hz]	μ [s${}^{-1}$]	ν [Hz]	μ [s${}^{-1}$]	ν [Hz]	μ [s${}^{-1}$]
120	114.3	10.05	103.9	11.62	98.1	13.20	93.4	14.98
140	84.2	7.36	74.6	8.71	72.5	9.66	69.2	11.02
160	65.8	5.40	57.6	6.52	55.6	7.70	53.1	8.02
180	52.2	4.52	48.2	5.05	44.3	6.01	42.0	6.82
200	42.4	3.42	38.6	4.31	34.8	4.62	34.1	5.52
220	36.4	2.89	30.5	3.22	29.0	3.82	28.4	4.33

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
