# Peer review of "A Study of Some Mechanical Properties of Composite Materials with a Dammar-Based Hybrid Matrix and Reinforced by Waste Paper"

_polymers, 2020, doi:10.3390/polym12081688_

Round 1

Reviewer 1 Report

This article reports the tensile properties and damping characteristics of hybrid dammar/epoxy resin-based composites that contain shredded and sheets of wastepaper. The article contains experimental characterization of these materials coupled with SEM images on the fracture surfaces and FTIR and EDS analysis.

Some of the results are interesting and would be of interest to the researchers working on the development of sustainable polymer processing and synthesis. I have several concerns about the content and the presentation that I detailed below. The authors need to restructure the figures and the manuscript to improve the readability of this article. I recommend another round of review after the article is revised.

How the resin and the shredded wastepaper mixed is not described? Was a mechanical mixer used to obtain a homogeneous mixture? What were the nominal size of the shredded paper pieces?  Details of the fabrication is important and affect to microstructure, thus need to be described better.

How did the authors ensure that the 18 layers of wastepaper sheets are uniformly distributed through the thickness (2.6 mm) of the laminate? Through-the-thickness SEM images of these laminates would have been very helpful to illustrate the microstructure and placement of these sheets within the laminate. What was the thickness of these wastepaper sheets?

Figure 4 can be removed as it does not depict anything important or original.

Figure 5a-c need to be combined into one figure to better illustrate the effect of the natural resin (dammar) ratio on the tensile response.

In Table 1, please give the average values of the properties with 90 or 95% confidence intervals, after the outliers are removed. Some researchers report average values and standard deviation, which is also acceptable. Giving the min and max values of the tests are not as informative for this case. Based on the results reported in Table 1, addition of dammar resin makes the resin mixture more ductile, while lowering the tensile strength and modulus.

Related to the comment above, the tensile properties of the neat epoxy without the natural resin should have been included in Table 1 as well as in Figure 5. One would be interested in knowing how the properties change compared to the baseline values of the neat resin. A minor point: Tensile modulus may be given in GPa.

Scale bars in Figure 8a-d and 12a-h are not clearly visible. Also, it seems that different magnifications are used in 8d. Barely readable scales bars in Figure 12 also point to different magnifications in some of the figures, which make comparison among these figures difficult.

I do not agree with the explanation given as to why a higher dammar natural resin ratio yields more voids in the cured laminate. There are several factors influencing the presence of the voids in the cured laminate. The authors suggest “As the polymerization was longer, the viscosity gradually increased until complete hardening. In this way, the bubbles that appeared during the reaction could no longer be removed and remained trapped in the resin.” When there is no dammar resin, the reaction and hardening (increase in viscosity) would be much faster, thus trapping more voids within the laminate which is the opposite of what is written by the authors. However, there are two other important factors governing the presence of voids that are not being addressed. One is the viscosity of the resin after the dammar is introduced. If mixing the dammar increased the viscosity of the resin system, then, although the cure reaction takes much longer, the higher viscosity slows down the removal of the voids. Another factor is how the mixing is carried out. During this phase, it may be possible that air bubbles are introduced into the mixture. It is common practice to degas the mixture after the hardener is mixed to remove these bubbles. There is no indication if this is done. Another possibility is the formation of the voids during cure, despite a void free mixture initially. These issues need to be articulated much better in the revised article.

The damping analysis of the resin and composites should be grouped together and presented under a separate subsection.

Figure 10a-d should be combined and given in one single Figure. Similarly, Figure 11a-d should be combined into a single figure as well.

Can the authors articulate what is expected if a much smaller percentage of natural resin, say 30% dammar, is used. The motivation of only using 60%, 70% and 80% dammar is not addressed in this article. What is the reason that these three ratios are selected?

Editorial comments:

The Abstract needs to be revised to enhance its readability.  Please include one or two sentences that summarize the most important results obtained. Please use “fracture surface” instead of “breaking surface”. Using “Moreover” and “also” in the same sentence is not needed.  The work that was carried out was summarized in the Abstract but there is no mention of the important or new findings.  One would be curious about the best ratio of the natural resin, or whether the mechanical properties were improved or not.

Introduction:

“… they can be manufactured at a low cost …” instead of “… they have a manufacturing low cost…”.

Please revised to: “The specific tensile strength and toughness make natural fibers an alternative to traditional reinforcements, …”.; “Also, natural fibers are easier to manipulate and require less energy during the processing, compared to glass or carbon fibers (see [7,8]).”

Please use: “elastic modulus” or “modulus of elasticity” not “elasticity modulus”

The sentence “The transversal tensile strength has an opposite tendency because the matrix ratio increased and the composite mixture becomes more uniform.” is not properly constructed. Hence, it is not clear what the authors are trying to convey. Please rewrite this sentence.

Please revise: “Larger investigations …” to “Comprehensive investigations…”

Please revise: “The studies made on these resins …” to “The studies on these resins …”

Please consider the following revision: “Dammar, the most commonly used natural resin, is harvested from the Dipterocarpaceae family trees of India and East Asia.”

Please avoid starting a paragraph or a new sentence with a reference number as was done several times on page 3.

Resoltech 1055 is often referred to as “the cure agent” or “hardener” not “the reinforcing agent” as was done on page 3.

The “characteristics curve” is better described as the “tensile response” of the composite or the neat polymer.

Page 14: please revise to: “The fact that the elongations at break of these composites are lower than the elongations at break of the hybrid resins …”

Author Response

Dear Reviewer 1,

Based on your comments, we have made the following changes to the structure of the paper with

Title: “A STUDY OF SOME MECHANICAL PROPERTIES OF COMPOSITE MATERIALS WITH A DAMMAR-BASED HYBRID MATRIX AND REINFORCED BY WASTE PAPER”

Authors: Marius Marinel Stănescu, Dumitru Bolcu*

Point 1: This article reports the tensile properties and damping characteristics of hybrid dammar/epoxy resin-based composites that contain shredded and sheets of wastepaper. The article contains experimental characterization of these materials coupled with SEM images on the fracture surfaces and FTIR and EDS analysis.

Some of the results are interesting and would be of interest to the researchers working on the development of sustainable polymer processing and synthesis. I have several concerns about the content and the presentation that I detailed below. The authors need to restructure the figures and the manuscript to improve the readability of this article. I recommend another round of review after the article is revised.

How the resin and the shredded wastepaper mixed is not described? Was a mechanical mixer used to obtain a homogeneous mixture? What were the nominal size of the shredded paper pieces?  Details of the fabrication is important and affect to microstructure, thus need to be described better.

Response 1: In terms of chemical structure, hybrid resins with various volumetric ratios of Dammar were studied in [Franz] (see the bibliography).

A document shredder was used to obtain the shredded paper, which cut the sheets of paper into 4 mm wide strips.

Then, the strips obtained were randomly cut into pieces with a length between 10-20 mm. A variable speed Steinhaus PRO-MX850 paint/mortar mixer was used to obtain a homogeneous mixture (resin-paper). An uniform pressure of 27000 N / m2 was applied to the plates.

Point 2: How did the authors ensure that the 18 layers of wastepaper sheets are uniformly distributed through the thickness (2.6 mm) of the laminate? Through-the-thickness SEM images of these laminates would have been very helpful to illustrate the microstructure and placement of these sheets within the laminate. What was the thickness of these wastepaper sheets?

Response 2: Sheets of paper with a thickness of 0.0105 mm and a specific mass of 80 g / m2 were used. The 18 layers of paper were pre-impregnated with resin. They were placed successively, one on top of the other, and at the end a uniform pressure of 27000 N / m2 was applied on the laminate. Figure 12a-d (becames Figure 10a-d after renumbering) shows sections of the breaking area that highlight the placement of the sheets in.

Point 3: Figure 4 can be removed as it does not depict anything important or original.

Response 3: Figure 4 has been removed.

Point 4: Figure 5a-c need to be combined into one figure to better illustrate the effect of the natural resin (dammar) ratio on the tensile response.

Response 4: To better illustrate the effect of the Dammar volume ratio on the tensile response of hybrid resins, the three graphs in Figure 5a-c were placed in a single figure (which became renumbered in Figure 4).

Point 5: In Table 1, please give the average values of the properties with 90 or 95% confidence intervals, after the outliers are removed. Some researchers report average values and standard deviation, which is also acceptable. Giving the min and max values of the tests are not as informative for this case. Based on the results reported in Table 1, addition of dammar resin makes the resin mixture more ductile, while lowering the tensile strength and modulus.

Response 5: Table 1 and Table 5 (after renumbering Table 4) entered the mean values and the corresponding standard deviations.

After Table 1, the observation regarding the ductility and, respectively, the decrease of the mechanical properties with the increase of the volume ratio Dammar was introduced.

Point 6: Related to the comment above, the tensile properties of the neat epoxy without the natural resin should have been included in Table 1 as well as in Figure 5. One would be interested in knowing how the properties change compared to the baseline values of the neat resin. A minor point: Tensile modulus may be given in GPa.

Response 6: The tensile properties of epoxy resin were not introduced in the text of the article because they are explicitly given by the manufacturer (see [Resoltech 1050] in the bibliography).

Point 7: Scale bars in Figure 8a-d and 12a-h are not clearly visible. Also, it seems that different magnifications are used in 8d. Barely readable scales bars in Figure 12 also point to different magnifications in some of the figures, which make comparison among these figures difficult.

Response 7: The images from Figure 8a-d (now renumbered in Figure 7) have been replaced with images with the same resolution.

Figure 12 was split in two (figure 10a-d and figure 11a-d after renumbering). The dimensions of the figures and their quality were increased.

Point 8: I do not agree with the explanation given as to why a higher dammar natural resin ratio yields more voids in the cured laminate. There are several factors influencing the presence of the voids in the cured laminate. The authors suggest “As the polymerization was longer, the viscosity gradually increased until complete hardening. In this way, the bubbles that appeared during the reaction could no longer be removed and remained trapped in the resin.” When there is no dammar resin, the reaction and hardening (increase in viscosity) would be much faster, thus trapping more voids within the laminate which is the opposite of what is written by the authors. However, there are two other important factors governing the presence of voids that are not being addressed. One is the viscosity of the resin after the dammar is introduced. If mixing the dammar increased the viscosity of the resin system, then, although the cure reaction takes much longer, the higher viscosity slows down the removal of the voids. Another factor is how the mixing is carried out. During this phase, it may be possible that air bubbles are introduced into the mixture. It is common practice to degas the mixture after the hardener is mixed to remove these bubbles. There is no indication if this is done. Another possibility is the formation of the voids during cure, despite a void free mixture initially. These issues need to be articulated much better in the revised article.

Response 8: The explanations have been modified according to your suggestions.

Based on the SEM analysis of the three types of hybrid resin we can see that the number of voids is higher when the Dammar volume ratio is higher. There are several factors influencing the presence of the voids in the cured laminate. One is the viscosity of the resin after the dammar is introduced. After mixing, the Dammar increased the viscosity of the resin system, then, although the cure reaction takes much longer, the higher viscosity slows down the removal of the voids. Another possibility is that voids form during the polymerization process. In the case of epoxy resin, the hardener produced a rapid reaction, the voids being removed in a short time, before the complete hardening of the resin (which occurred in less than 24 hours). When the epoxy resin was mixed with Dammar natural resin then the hardening time is much longer (72-96 hours). In this way, due to the high viscosity, the voids that appeared during the reaction could no longer be removed and remained trapped in the resin.

Point 9: The damping analysis of the resin and composites should be grouped together and presented under a separate subsection.

Response 9: A separate section on the vibrations of the studied resins and composites was presented.

Point 10: Figure 10a-d should be combined and given in one single Figure. Similarly, Figure 11a-d should be combined into a single figure as well.

Answer 10: The tensile response of the composite materials in Figure 10a-d (now renumbered in Figure 8) was combined into a single figure. The procedure was similar to Figure 11a-d (now renumbered Figure 9).

Point 11: Can the authors articulate what is expected if a much smaller percentage of natural resin, say 30% dammar, is used. The motivation of only using 60%, 70% and 80% dammar is not addressed in this article. What is the reason that these three ratios are selected?

Response 11: The name hybrid resin was given after finding the method of polymerizing the natural resin Dammar with the help of epoxy resin and the related hardener.

"Mixtures" with different volume ratios of natural resin Dammar, respectively epoxy resin with the corresponding hardener were obtained and tested (in terms of mechanical properties and chemical structure) during 5 years of research. However, from the beginning there was a desire to obtain a hybrid resin as "bio-degradable". To achieve this goal, the problem was to find the maximum volume ratio of natural resin in the composition of the hybrid resin so that it does not lose its elastic properties. It was observed experimentally that if the volume ratio of Dammar increases over 80% (even by a few percent), the polymerization process is very long and the hybrid resin obtained is not of interest in terms of mechanical properties. Based on the above observation, the volume ratios of Dammar of 60%, 70% and 80% respectively were chosen in this article.

Editorial comments:

Point 12: The Abstract needs to be revised to enhance its readability.  Please include one or two sentences that summarize the most important results obtained. Please use “fracture surface” instead of “breaking surface”. Using “Moreover” and “also” in the same sentence is not needed.  The work that was carried out was summarized in the Abstract but there is no mention of the important or new findings.  One would be curious about the best ratio of the natural resin, or whether the mechanical properties were improved or not.

Introduction:

“… they can be manufactured at a low cost …” instead of “… they have a manufacturing low cost…”.

Response 12: The summary of the article has been revised.

The phrase "breaking surface" was replaced by "fracture surface";

The phrase “they have a low cost manufacturing” has been replaced by “they can be manufactured at a low cost”.

Point 13: Please revised to: “The specific tensile strength and toughness make natural fibers an alternative to traditional reinforcements, …”.; “Also, natural fibers are easier to manipulate and require less energy during the processing, compared to glass or carbon fibers (see [7,8]).”

Response 13: The indicated text has been revised.

Point 14: Please use: “elastic modulus” or “modulus of elasticity” not “elasticity modulus”

Answer 14: It was replaced in the text of the article "elasticity modulus" with "modulus of elasticity".

Point 15: The sentence “The transversal tensile strength has an opposite tendency because the matrix ratio increased and the composite mixture becomes more uniform.” is not properly constructed. Hence, it is not clear what the authors are trying to convey. Please rewrite this sentence.

Response 15: The indicated sentence has been reformulated.

Point 16: Please revise: “Larger investigations …” to “Comprehensive investigations…”

Response 16: The indicated text has been revised.

Point 17: Please revise: “The studies made on these resins …” to “The studies on these resins …”

Response 17: It has been replaced “The studies made on these resins…” to “The studies on these resins…”

Point 18: Please consider the following revision: “Dammar, the most commonly used natural resin, is harvested from the Dipterocarpaceae family trees of India and East Asia.”

Response 18: The requested change has been made.

Point 19: Please avoid starting a paragraph or a new sentence with a reference number as was done several times on page 3.

Response 19: The sentences that started with the bibliographic reference were reformulated.

Point 20: Resoltech 1055 is often referred to as “the cure agent” or “hardener” not “the reinforcing agent” as was done on page 3.

Response 20: "The reinforcing agent" was replaced with "hardener".

Point 21: The “characteristics curve” is better described as the “tensile response” of the composite or the neat polymer.

Response 21: It has been replaced in the text of the article “characteristic curve” to “tensile response”.

Point 22: Page 14: please revise to: “The fact that the elongations at break of these composites are lower than the elongations at break of the hybrid resins …”

Response 22: The requested change has been made.

We mention that the answers addressed to reviewer 1 are colored in the text of the paper in blue, and the answers addressed to both reviewers are colored in brown.                                                                                                    

Thanks for the views expressed on the basis of which we have made the changes that have contributed to increasing the scientific level of the paper.

                                                                                     Authors

Reviewer 2 Report

The authors of the submitted manuscript investigated the influence of addition of various forms of cellulose as the reinforcing material and Dammar as the matrix materials and mechanical and physical performance of the resulting composites. The authors did not provide clear goal of this study. It was not mentioned what was the purpose of the study and fields of application of the resulting material. The obtained results, especially from the vibration tests raise some doubts (see comments below). In this light, the final conclusions are also questionable. I recommend this manuscript for the major revision, giving a chance to the authors to answer the comments and explain pointed deficiencies.

1) The abstract need to be corrected. The authors use personal form and the most of sentences there starts from “we”. The impersonal form would look like much better and it is a kind of standard in scientific writing. This is also applicable to the whole manuscript.

2) Page 1, line 24: “vinyl polychloride” – the nomenclature “polyvinyl chloride” is usually used for description of this material. Suggested to change.

3) In the second paragraph of the Introduction the authors wrote: “The specific tensile strength

and toughness make natural fibers be an alternative to traditional reinforcements, such as glass fibers or other filling materials.”, however, the mechanical properties of natural fibers are usually much worse than traditional reinforcing materials, in general, even to the glass fiber. Please address to this comment.

4) It is suggested to join the paragraphs in lines 73-82 into a single one.

5) If a sentence starts with a reference, it is recommended to add “Ref.” at the beginning.

6) The Introduction is somehow chaotic. The authors referred to numerous studies, but without any analysis and interconnection between cited references. The goals and originality of the manuscript are not appropriately emphasized. It is suggested to reorganize the Introduction in such a way to emphasize the goals of the study as well as its originality with respect to cited references.

7) It is suggested to rename section 2.1 to “Specimens manufacturing” or similar.

8) Please discuss what was the reason that the specimens were of different thickness.

9) Please also discuss on which basis the content of Dammar was selected for tests. Did the authors performed any preliminary studies which allowed to select the content of 60%, 70%, and 80%?

10) The experimental setups, in particular, for vibration damping need to be presented and described in the manuscript.

11) According to the discussion in lines 181-188, addition of Dammar influences negatively on mechanical properties of the resulting material. Please comment on this.

12) What was the purpose of re-testing the same specimen using FTIR? Why the results of FTIR tests for other considered specimens were not presented and discussed?

13) The results presented in Fig. 7 are not readable. The authors should scale longitudinal axis to 5 keV, since no relevant information is presented after this value. The plot should be also enlarged.

14) In vibration tests, no information on excitation was given. The plots presented in Figs. 9,13,14 are difficult to follow. Firstly, all the descriptions should be presented in English. Secondly, data representing the curves should be exported and plotted separately from the interface of the acquisition software.

15) The information presented in Figs. 9, 13, 14 is questionable. There is no direct answer in the manuscript on how the authors determined damping factors.

16) The equation after Fig. 9 (not numbered) is valid for linear elastic materials, while the authors showed earlier that their materials are non-linear.

17) Presenting results in Table 4 is questionable. It is obvious that the parameters such as frequency changes with a change of the length of a beam. While the damping of material is, in fact, depends on frequency. Moreover, in the case of viscoelastic materials it is usually given in terms of the loss factor.

18) All the microphotographs need to be enlarge to make them readable, including the legends.

19) It is not clear how the authors selected testing methods and what was their purpose in this study.

20) The manuscript contains numerous grammar and stylistic errors. The proofreading is highly recommended. Some sentences, like “We observed ASTM E168 and E1252 (see [49,50]).” or “We observed ASTM F2603 (see [52])." are unclear.

Author Response

Dear Reviewer 2,

Based on your comments, we have made the following changes to the structure of the paper with

Title: “A STUDY OF SOME MECHANICAL PROPERTIES OF COMPOSITE MATERIALS WITH A DAMMAR-BASED HYBRID MATRIX AND REINFORCED BY WASTE PAPER”

Authors: Marius Marinel Stănescu, Dumitru Bolcu*

Point 1: The abstract need to be corrected. The authors use personal form and the most of sentences there starts from “we”. The impersonal form would look like much better and it is a kind of standard in scientific writing. This is also applicable to the whole manuscript.

Response 1: The summary has been modified. The impersonal form was used throughout the article.

Point 2: Page 1, line 24: “vinyl polychloride” – the nomenclature “polyvinyl chloride” is usually used for description of this material. Suggested to change.

Response 2: "Vinyl polychloride" was replaced by "polyvinyl chloride".

Point 3: In the second paragraph of the Introduction the authors wrote: “The specific tensile strength and toughness make natural fibers be an alternative to traditional reinforcements, such as glass fibers or other filling materials.”, however, the mechanical properties of natural fibers are usually much worse than traditional reinforcing materials, in general, even to the glass fiber. Please address to this comment.

Response 3: A comment has been added to your comment in the text of the article

Point 4: It is suggested to join the paragraphs in lines 73-82 into a single one.

Response 4: The two paragraphs have been merged.

Point 5: If a sentence starts with a reference, it is recommended to add “Ref.” at the beginning.

Response 5: The sentences that started with the bibliographic reference were reformulated.

Point 6: The Introduction is somehow chaotic. The authors referred to numerous studies, but without any analysis and interconnection between cited references. The goals and originality of the manuscript are not appropriately emphasized. It is suggested to reorganize the Introduction in such a way to emphasize the goals of the study as well as its originality with respect to cited references.

Response 6: Some clarifications have been made on the structure of the introduction.

It was desired to obtain composite materials in which the matrix was made of hybrid resin based on Dammar, and the reinforcer was made of waste paper. In the first phase of the introduction of the article, studies were presented that investigated the properties of natural fibers (which contain a significant amount of cellulose). Then, studies were cited that showed the properties of cellulose. Subsequently, studies on natural resins were presented, especially those that investigated the properties of Dammar resin. At the end of the introduction, studies of composite materials with Dammar resin matrix and natural fiber reinforcement were cited.

Point 7: It is suggested to rename section 2.1 to “Specimens manufacturing” or similar.

Response 7: Subsection 2.1 has been renamed “Specimens manufacturing”.

Point 8: Please discuss what was the reason that the specimens were of different thickness.

Response 8: The hybrid resin was initially studied. In order for the resin specimens to be able to withstand the tensions of the tensile test machine, it was necessary for them to have a minimum thickness of 6 mm. It was ensured that all the plates cast in hybrid resin had the same thickness. For sheet-reinforced composites, we chose a number of 18 sheets of recycled paper because their total area was 1 m2 and their mass was 80 g. The thickness of the laminates obtained was 2.6 mm. In the case of composites reinforced with recycled shredded paper, a set of 2.6 mm thick plates was initially poured. The tensile test found that the test pieces broke when they were caught in the tanks of the test machine. Consequently, the thickness of the re-cast plates with this type of reinforcement was increased to 5.2 mm (twice the thickness of the initially cast). To ensure the homogeneity of the mixture it was necessary to increase the proportion of hybrid resin.

Point 9: Please also discuss on which basis the content of Dammar was selected for tests. Did the authors performed any preliminary studies which allowed to select the content of 60%, 70%, and 80%?

Response 9: The name hybrid resin was given after finding the method of polymerizing the natural resin Dammar with the help of epoxy resin and the related hardener.

"Mixtures" with different volume ratios of natural resin Dammar, respectively epoxy resin with the corresponding hardener were obtained and tested (in terms of mechanical properties and chemical structure) during 5 years of research. However, from the beginning there was a desire to obtain a hybrid resin as "bio-degradable". To achieve this goal, the problem was to find the maximum volume ratio of natural resin in the composition of the hybrid resin so that it does not lose its elastic properties. It was observed experimentally that if the volume ratio of Dammar increases over 80% (even by a few percent), the polymerization process is very long and the hybrid resin obtained is not of interest in terms of mechanical properties. Based on the above observation, the volume ratios of Dammar of 60%, 70% and 80% respectively were chosen in this article.

Point 10: The experimental setups, in particular, for vibration damping need to be presented and described in the manuscript.

Response 10: The experimental settings made for vibration recording are presented in subsection 2.2.3.

Point 11: According to the discussion in lines 181-188, addition of Dammar influences negatively on mechanical properties of the resulting material. Please comment on this.

Answer 11: This phenomenon can be explained by the fact that the resins, even in solid state, have a rheological behavior characterized by viscosity. Due to external stresses, a low viscosity leads to significant deformations. Dammar resin diluted with turpentine remains liquid, so it has a low viscosity. Solidification occurs by mixing with synthetic resin. After curing, the resulting mixture will have a rheological behavior whose viscosity decreases with increasing volume of Dammar. Therefore, at the same external load, the deformations of the hybrid resin by 80% Dammar are higher than the deformations of the hybrid resin by 70%, respectively 60% volume ratio of Dammar.

Point 12: What was the purpose of re-testing the same specimen using FTIR? Why the results of FTIR tests for other considered specimens were not presented and discussed?

Response 12: In-depth studies on the composition and chemical properties of Dammar-based hybrid resin (with various volume ratios) were performed in [Franz] in the bibliography. The hybrid resin with a volume ratio of 70% Dammar (from the point of view of the chemical structure) was studied using the FTIR analysis, because it falls within the average of the chemical properties of the three types of hybrid resin. A hybrid resin sample with a volume ratio of 70% Dammar was re-tested to show that the results obtained at the first test were not flawed.

Point 13: The results presented in Fig. 7 are not readable. The authors should scale longitudinal axis to 5 keV, since no relevant information is presented after this value. The plot should be also enlarged.

Response 13: The quality (clarity) of figure 7 has been improved (now renumbered figure 6).

Point 14: In vibration tests, no information on excitation was given. The plots presented in Figs. 9,13,14 are difficult to follow. Firstly, all the descriptions should be presented in English. Secondly, data representing the curves should be exported and plotted separately from the interface of the acquisition software.

Response 14: For vibration measurement, each bar was embedded at one end and the measuring transducer was mounted at the free end. The bar was deformed by applying a force to the free end. The free vibrations obtained by eliminating the force specified above were measured. The clarity of figures 9, 13, 14 has been improved (now figures 12, 13, 14 have been renumbered).

Point 15: The information presented in Figs. 9, 13, 14 is questionable. There is no direct answer in the manuscript on how the authors determined damping factors.

Response 15: At the recording in figure 9 (became figure 12 after renumbering) for the determination of the damping factor, the first marker (red dotted line in the graph) was placed on the maximum with the value w1 = 1.9111, corresponding to the time moment t1 = 1.5413 and the second marker (blue dotted line in the graph) on the maximum with the value w2 = 1.1046, corresponding to the time moment t2 = 1.6150. The damping factor was calculated with the formula (3.3.1). The damping factor was similarly calculated for all experimental measurements.

Point 16: The equation after Fig. 9 (not numbered) is valid for linear elastic materials, while the authors showed earlier that their materials are non-linear.

Response 16: During the free vibrations, which have been measured, the stresses that occur are small and therefore the materials from which the bars are made are required in the elastic field, where there is proportionality between specific stresses and strains (in this area Hooke's law is valid). The depreciation coefficient formula was numbered.

Point 17: Presenting results in Table 4 is questionable. It is obvious that the parameters such as frequency changes with a change of the length of a beam. While the damping of material is, in fact, depends on frequency. Moreover, in the case of viscous-elastic materials it is usually given in terms of the loss factor.

Response 17: The natural frequencies of the bars depend on not only the dimensions (thickness, length), but also on the material properties (density and modulus of elasticity). Because the test samples studied had similar dimensions, the differences between the measured frequencies are due to the differences between the modulus of elasticity of the composite materials from which the samples are made. We found that there is a correspondence between the measured frequencies and the modulus of elasticity determined at the tensile test, more precisely the increase of the modulus of elasticity leads to the increase of the vibration frequency. The damping factor as a whole characterizes the vibration damping capacity for a test piece. We also found that the vibration damping capacity increases with increasing volume ratio of Dammar in the hybrid resin used as a matrix. Indeed, the vibration damping capacity for materials is given by the loss factor. For the studied materials the values of the loss factor are presented in the section “Discussions and conclusions”.

Point 18: All the microphotographs need to be enlarge to make them readable, including the legends.

Response 18: The images from Figure 8a-d (now renumbered in Figure 7) have been replaced with images with the same resolution.

Figure 12a-h was split in two (figure 10a-d and figure 11a-d after renumbering). The dimensions of the figures and their quality were increased.

Point 19: It is not clear how the authors selected testing methods and what was their purpose in this study.

Response 19: For hybrid resins and manufactured composite materials were studied:

- mechanical properties (modulus of elasticity, tensile strength, elongation at break and tensile response) - using tensile stress;

- fracture area imaging - using SEM analysis;

- chemical composition - using FTIR and EDS analyzes;

- free vibration behavior.

It was proposed to find new possibilities for the use of paper waste and to determine the limits of applicability of composites with this type of reinforcer. Based on the determined properties and to use these composite materials in the medical field or in the automotive industry.

Point 20: The manuscript contains numerous grammar and stylistic errors. The proofreading is highly recommended. Some sentences, like “We observed ASTM E168 and E1252 (see [49,50]).” or “We observed ASTM F2603 (see [52])." are unclear.

Response 20: Grammatical and stylistic errors have been corrected. For example, we replaced "We observed ASTM E168 and E1252 (see [49,50])." cu “Have been met the requirements of the ASTM E168 and ASTM E1252 standards (see [49,50]).”

We mention that the answers addressed to reviewer 2 are colored in the text of the paper in green, and the answers addressed to both reviewers are colored in brown.

Thanks for the views expressed on the basis of which we have made the changes that have contributed to increasing the scientific level of the paper.

                                                                                                                Authors
